# Esports experts have a wide gaze distribution and short gaze fixation duration: A focus on League of Legends players

**Inhyeok Jeong**[1], **Kazutoshi Kudo**[1,2], **Naotusgu Kaneko**[1], **Kimitaka Nakazawa**[1] *

**1** Department of Life Sciences, Graduate School of Arts and Sciences, The University of Tokyo, Tokyo, Japan, **2** Graduate School of Interdisciplinary Information Studies, The University of Tokyo, Tokyo, Japan

* nak_kmtk@idaten.c.u-tokyo.ac.jp

**Data Availability Statement:** All relevant data are within the paper and its Supporting Information files.

## Abstract

This study investigated the specific gaze control ability of expert players and low-skill players of League of Legends (LoL). Eleven expert and nine low-skill players were divided according to their official ranking. Then, the gaze movement of each participant when performing each task (e.g., easy task and moderate task) while competing against a computer artificial intelligence system was recorded. Experts were found to have a significantly wide horizontal gaze distribution. Additionally, experts had a consistently short gaze fixation duration during the moderate task. These results suggest that a wide horizontal gaze distribution allows experts to obtain information from a wider area. Additionally, the consistently short fixation duration of the experts indicated that they need only a short period to assess information, which is advantageous because large amounts of information need to be processed within a limited time while playing. This specific gaze control ability could be an important factor that contributes to the superior performance of expert LoL players.

## Introduction

Esports, otherwise known as electronic sports, consist of competitive video gaming and are performed by numerous offline and online audiences [1]. Since 2015, publications about esports have been consistently increasing [2], and researchers have focused on the many various aspects of esports.

Esports competitions are mainly performed online (i.e., using a computer monitor) and require optimal cognitive, physical, and mental abilities [3]. Therefore, many researchers have been assessing and analyzing the cognitive function of esports players. As a first step, researchers have started to evaluate the effects of video games. For example, the players of first-person shooting (FPS) games, which comprise a popular video game genre, exhibit quick decision-making, strategizing, and reaction times [4]. Both single case studies and meta-analysis studies have discovered that many video games increase the various functions of cognitive performance, such as reaction time and problem-solving ability [5]. Additionally, compared to non-players, video game players are known to have a large-capacity visual working memory [6].

Although some previous research of video games has been conducted, research of the characteristics of esports players has been insufficient. Esports differ from simple video games

**Funding:** This research is supported by the JSPS KAKENHI (grant numbers JP18H04082 and JP18HKK0272), JST-Mirai Program (grant number JP20349063), and JST-MOONSHOT program (grant number JPMJMS2012–2-2–2). the funders had no role in study design, data collection and analysis, decision to publish, or preparation of the manuscript.

**Competing interests:** The authors have declared that no competing interests exist.

because they include a competitive element. Pedraza-Ramirez et al. indicated that not all video games are esports games, and they suggested that research of the effects of esports on cognitive function is lacking [7]. Recently, Leis and Lautenbach [8] argued that competitive esports can affect physiological stress. The cognitive effects of esports and behavioral characteristics of esports players are unknown as well. It is possible that specific gaze movement is a key element involved in winning the game [9–11]. In this article, gaze movement refers to the movement of a visual pivot point. Almeida et al. indicated that analyses of gaze movement using the eye-tracking method are generally applied in many fields of study [12]. Additionally, analyses of gaze movement are useful for understanding the perceptual-cognitive processing involved in athletic performance [13]. One research study that analyzed gaze movement during esports found out that highly skilled esports (real-time strategy games) players perform specific saccadic gaze movements [9]; however, that study did not evaluate the fixation characteristics present while playing esports. Additionally, the difference between the fixation gaze movement of professional esports players and non-professional esports players was compared by observing players of a soccer simulation video game called FIFA 19 [10]. Despite focusing on the fixation of gaze movement, no significant difference in the duration of the fixation time and the number of fixations of the professional esports players and non-professional esports players was observed. However, it was found that professional FPS players can respond more quickly to visual stimuli than non-professional FPS players [11].

After comparing the results of three research studies by Jeong et al. [9], Bickmann et al. [10], and Koposov et al. [11], we found that various esports genres require specific gaze control strategies. However, the characteristics of gaze movement, especially fixation on the multiplayer online battle arena (MOBA), remain unknown. Gaze movement, especially fixation, has an important role in visual information processing because most of the visual experience is created during fixation [14]. In other words, analyses of gaze movement characteristics, especially fixation, can be helpful to understanding gaze movement of esports players and can help determine the origins of the high-performance ability of esports experts. Therefore, we designed this study using the League of Legends (LoL) video game to accurately represent the MOBA game genre [15]. LoL was developed by Riot Games in 2009 (https://www.leagueoflegends.com/). The total number of players exceeds 70 million, and it is one of the most popular games in the MOBA game genre [15]. Players have to properly control and level-up their character to win the game. Specifically, players have to collect virtual commodities (called "gold") that are used to kill their enemy characters and increase their skill level. During this process, both a fast reaction time and correct motor control ability are required. Specifically, during LoL, players are required to have good motor control ability to accurately operate a mouse while pressing the appropriate button on a keyboard as quickly as possible. While playing LoL, accurate gaze movements are required to immediately identify the enemy's characteristics and respond accordingly. Confirmation of the information through gaze movement is the first step involved in information processing using cognitive functions [16]. Participants have to respond immediately and eliminate as many enemies as possible within the set time of a task. Thus, to achieve the best performance within the given time, it is necessary to process as much visual information as possible within a short time; while playing LoL, this information is provided on a wide screen. If LoL players can obtain a large amount of visual information provided over a wide area within a short period of time, then they can shorten the information-processing time. This is essential for esports because quick reactions and decisions are required to be successful. During esports, it is beneficial to have shorter information-processing times than those of the opponent players.

During this study, we assumed that highly skilled LoL players (experts) have a wider gaze distribution and shorter gaze fixation duration than LoL players with lower skill levels.

## Methods

### Participants

All study participants were recruited through social media. Before beginning the experiment, we performed a power analysis to estimate the required sample size (G*Power, version 3.1.9). The power analysis was conducted using the moderate task performance level of expert players (n = 4) and low-skill players (n = 4) exhibited during a preliminary experiment (Cohen's d: 1.4; α level: 0.05; power [1-β error probability]: 0.8). Cohen's d was calculated using Cohen's method [17]. The specific protocol of G*power is available in the S2 File. To account for any possible data losses, 11 expert and 10 low-skill LoL players were recruited for the experiment. All participants who performed the task had normal or corrected-to-normal vision.

Data of one participant in the low-skill player group were discarded because of poor quality. Finally, we analyzed the data of 20 participants with experience playing LoL (11 expert and 9 low-skill players). The mean age of the expert players was 22.6 years (standard deviation [SD], 2.6 years). The mean age of the low-skill players was 20.2 years (SD, 1.5 years). Expert and low-skill players were divided based on their official rank. Iron, bronze, and silver ranks were assigned to low-skill players, whereas gold, platinum, and diamond ranks were assigned to expert players. Low-skill players had an average of 2 years (SD, 1.9 years) of playing experience; however, experts had an average of 8.6 years (SD, 2.0 years) of playing experience. This difference in experience was significant (t = -4.83; p < .001). The characteristics and background information of each participant are shown in Table 1. Before the experiment was conducted, we verbally provided details about the experiment to the players. Then, all participants were asked to provide their written consent by signing the consent form. The written consent form included the purpose of the study, methods, privacy policy, and risks of the experiment; the

**Table 1. Participant characteristics and basic information.**

| Participants, n | |
|---|---|
| Expert | 9 |
| Low-skill | 11 |
| **Age, yr** | |
| Expert | 20.2±1.5 |
| Low-skill | 22.6±3.5 |
| **Sex, female/male, n** | 0/20 |
| **Dominant hand, right/left, n** | 19/1 |
| **Education level, n** | |
| High school | 16 |
| Undergraduate | 2 |
| Graduate | 2 |
| **LoL experience, yr** | |
| Expert | 8.6±2.0 |
| Low-skill | 2.0±1.9 |
| **Highest LoL rank, n** | |
| Iron | 6 |
| Bronze | 1 |
| Silver | 2 |
| Gold | 6 |
| Platinum | 4 |
| Diamond | 1 |

option to withdraw consent was also provided. This research was approved by the Human Research Ethics Committee of the University of Tokyo (Institutional Review Board: Ethical Review Committee for Experimental Research Involving Human Subjects, Graduate School of Arts and Sciences and the College of Arts and Sciences, University of Tokyo). All data were collected after the Human Research Ethics Committee of the University of Tokyo approved this research (August 25, 2022 to September 3, 2022).

After all participants signed the consent form, they completed a questionnaire about their esports playing experience and skill level. According to the questionnaire, the expert group played an average of 2 (SD, 1.5) different games other than LoL, and the low-skill group played an average of 1.3 games (SD, 0.7) different games other than LoL. This difference was not significant. Additionally, participants were instructed to use a scale of 1 to 10 to rank the skill level of their character (described in the Task section) used during the task. Expert players had more knowledge when playing as specific characters used during the task than low-skill players (unpaired t-test; expert players: $5.90 \pm 2.4$; low-skill players: $1.33 \pm 1.8$; $p < .001$). To perform comparisons with previous research of LoL, the Edinburgh Handedness Inventory was used to determine which hand was preferred by the players [18, 19]; only one participant preferred the left hand. However, all participants operated the mouse with the right hand and the keyboard with the left hand.

## The task

The task was performed using the LoL custom play mode. Before starting the task, we instructed the participants to destroy as many enemies as possible. Easy and moderate tasks were performed for 12 minutes, resulting in a total of 24 minutes of playing. The overall features of the task are detailed in Fig 1. To consider the features of each character, Master Yi, Garen, and Galio were used during the experiment. First, all participants used the same character (Master Yi) (Fig 1A) to complete the task. After selecting the character, all participants were required to use the same item, skill, and spell to complete the task. The character was moved by right-clicking the mouse. All participants were competing against a computer artificial intelligence (AI) system in the same place (Bottom lane; Fig 1E). Additionally, the same characters, namely, Garen (Fig 1B) and Galio (Fig 1C), were present. To destroy the enemy units, the participant had to use skill (combinations of Q, W, E, and R keys) and normal (left-clicking the mouse) attacks. To use stronger skills, participants first had to destroy Minions (Fig 1D) and collect the virtual commodity (gold). The procedure for the easy and moderate tasks remained the same. However, the AI during the moderate task had higher-level intelligence. During the moderate task, the AI system collected gold 1.5-times faster than during the easy task, and skills were used more frequently. Therefore, participants automatically experienced a higher difficulty level while performing the moderate task.

## Experimental procedure

All participants sat in front of the monitor and wore the Pupil Labs eye tracker (Pupil Labs UG Haftungsbeschränkt, Berlin, Germany). We measured the distance (100 cm) between each participant and the monitor. Then, we asked the participants to maintain the same head position that they used during the start of the task. The easy and moderate tasks were each performed for 12 minutes. All participants completed the tasks during a single session. To avoid the order effect, the order of the tasks was counterbalanced. During the task, gaze movement was recorded using Pupil-Capture software (version 3.5; https://github.com/pupil-labs/pupil/releases/tag/v3.5). The task and recording ended after 12 minutes. After finishing the task, the number of destroyed enemy units was recorded.

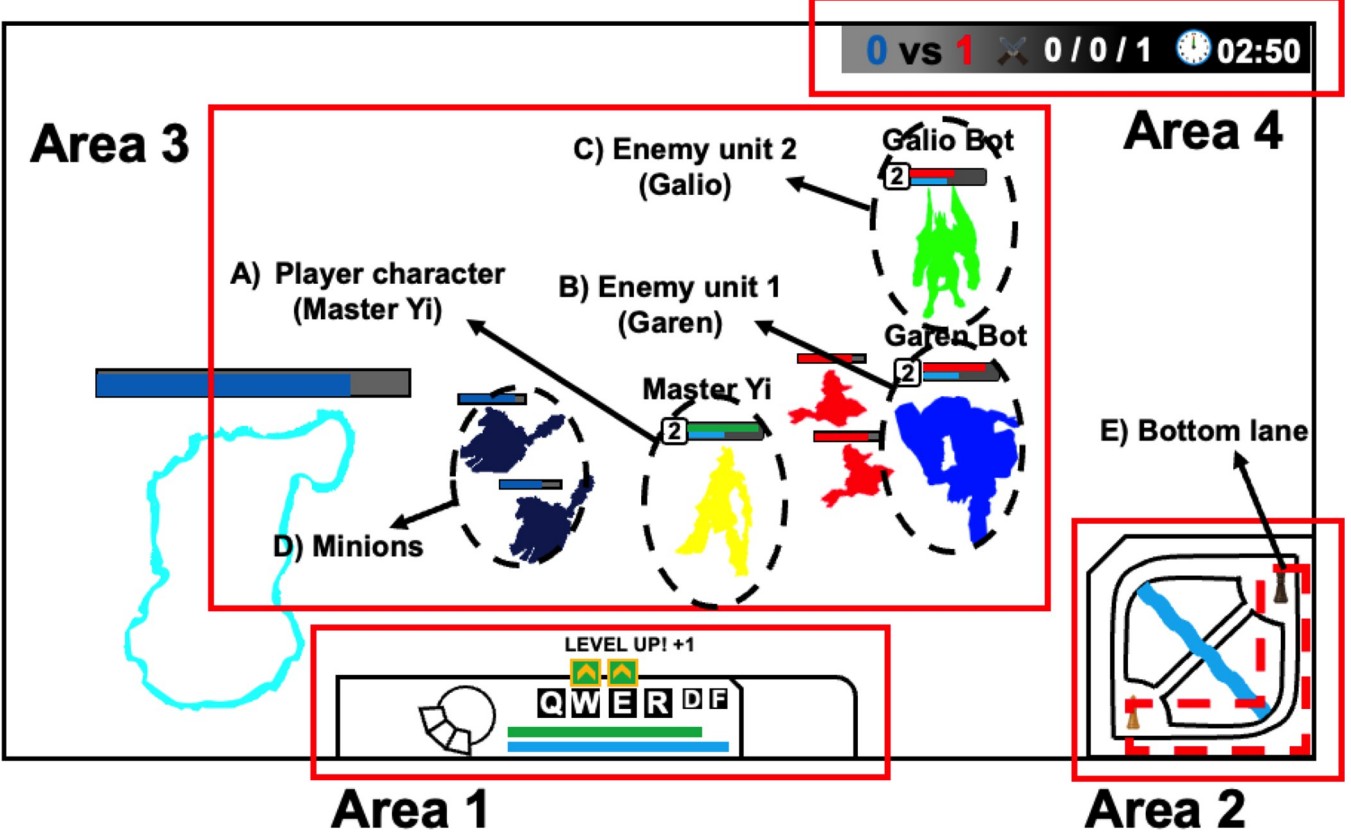

**Fig 1. Approximate play scene of the task.** Area 1: status of the character controlled by participants (i.e., skill cooldown time and remaining health points). Area 2: mini-map showing the overall flow of the task. Area 3: main play area of the task. Area 4: play time and score.

### Equipment

The experiment utilized a 24-inch × 32-inch monitor. Participants were allowed to use their monitor of choice. During the task, gaze movement was collected by the Pupil-Core (Pupil Labs UG Haftungsbeschränkt). Pupil-Core was validated by the manufacturer [20]. Pupil-Capture version 3.5 (gaze movement recording software) was used to collect gaze movement. Gaze movement was measured using one field camera (60 Hz at 1910 × 1080 pixels) and two eye cameras (200 Hz at 192 × 192 pixels). A field camera was used to record the experimental environment and calibrate the gaze movement using the Screen Marker Calibration method [20]. After calibration, four different markers (width × height: 4 cm × 4 cm) were attached to the monitor (Fig 2A) to normalize the gaze movement, which was quantified using a value between 0 and 1 based on the actual monitor size. After recording the gaze movement, Pupil-Player version 3.5 software and a 2022 MacBook Pro 16 (Apple Inc., Cupertino, CA, USA) were used to visualize and identify the output of the gaze movement. The gaze position was visualized using a green dot (Fig 2B) in the exported video. Then, the gaze movement data were exported to an Excel file (Microsoft Corporation, Redmond, WA, USA).

### Gaze movement analysis

All gaze movement data were exported using Pupil-Player version 3.5 software. Gaze movement data with a low confidence level (less than 60%) were automatically removed. The preparation time after the start of the task was set to 1.5 minutes; all gaze movement data gathered

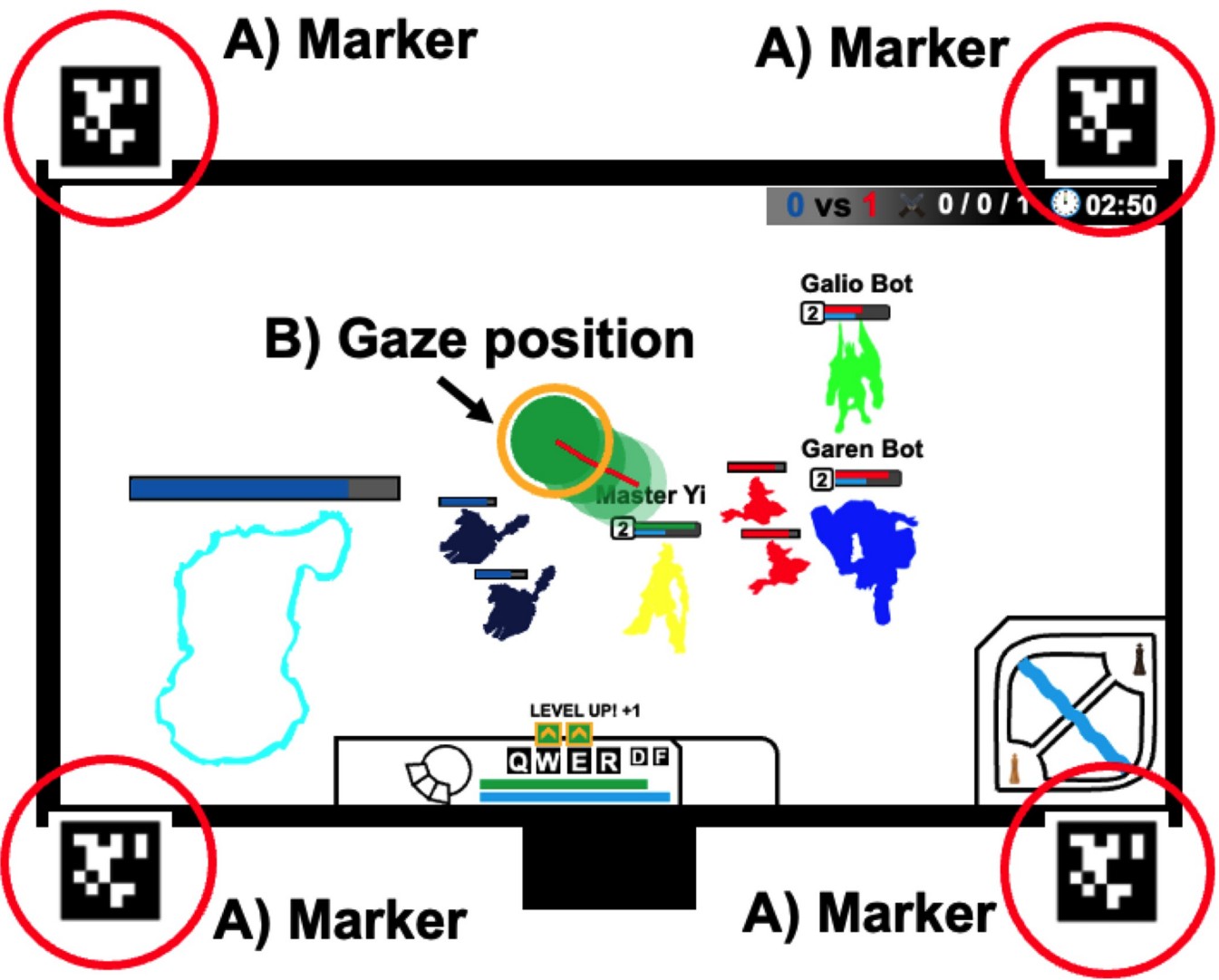

**Fig 2. Normalization of the gaze movement.** **(A)** Four markers are attached to the monitor. **(B)** Example of a gaze position.

during this time were not used for the analysis. Because each participant used different monitors of various sizes, these data were normalized based on the monitor size (mm) used. Specifically, the coordinate position of the gaze was specified according to the size of the monitor and expressed as a number between 0 to 1. Then, the SD of the gaze movement (horizontal and vertical directions) of all participants was separately calculated during each trial and subsequently averaged (within-participant SD). Moreover, the center of the heatmap, which indicates the density of gaze points and the center of the screen, was calculated. Fixation was identified when the gaze movement was fixed at 100 ms or more and the maximum pupil dispersion was less than 1.5 degrees. The location of each fixation for each area was designated as an area of interest (AOI) based on the task interface used during the current experiment (Fig 1). Additionally, the duration of the fixation time, SD of the fixation time, and number of fixations were calculated. To calculate the SD of the fixation time, the SD of each participant's fixation duration during each trial was calculated; then, the average of these values was expressed (within-participant SD). Additionally, data acquired while the participant's character was dead were not used for the analysis.

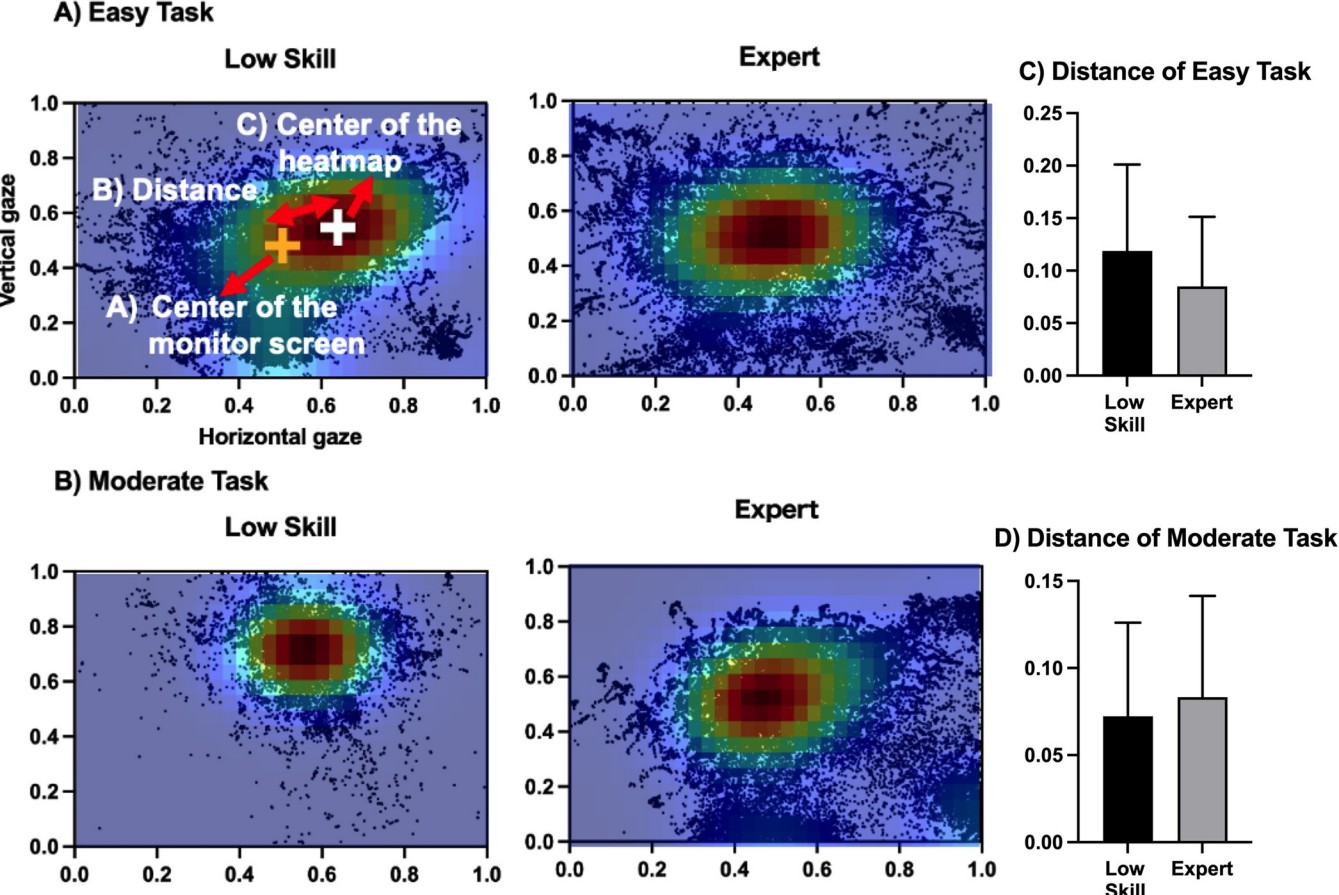

**Fig 3. Example of the gaze movement and the distance between the center of the monitor screen and the center of the heatmap.** (A and B) Examples of gaze movement during the task. Each black dot shows the location of the gaze movement on the monitor. The heatmap shows the density of the gaze position (red: higher density of the gaze position; blue: lower density of the gaze position. (A) Center of the monitor screen. (B) Distance between the center of the heatmap and the center of the monitor. (C) Center of the heatmap. The values of the x-axis and y-axis are calibrated using a number between 0 and 1 based on the actual monitor size. (C and D) The average distances (center of the monitor and center of the heatmap) of the expert and low-skill players. The bar shows the average distance (center of the monitor and center of the heatmap) of expert and low-skill players. The error bars represent the standard deviation (SD) of each value.

## Performance level analysis

During this experiment, the performance level was defined as the kill-to-death ratio. To compare the performance level of expert and low-skill players, the number of enemy (Garen and Gallio) kills and the number of times the player's character died during the easy and moderate tasks were used to determine the performance level. Additionally, the difference between the easy task performance level and moderate task performance level was analyzed to determine whether the task difficulty was appropriately set.

## Statistical analysis

All statistical analyses were conducted using RStudio version 2022.02.1+461 (RStudio, Boston, MA, USA). The homogeneity of variance and normality of variance were identified using Levene's test and Shapiro-Wilk's test, respectively. The results showed that the performance level of the moderate task, AOI, fixation durations of the easy and moderate tasks, number of fixations of the easy and moderate tasks, and the distance between the center of the heatmap

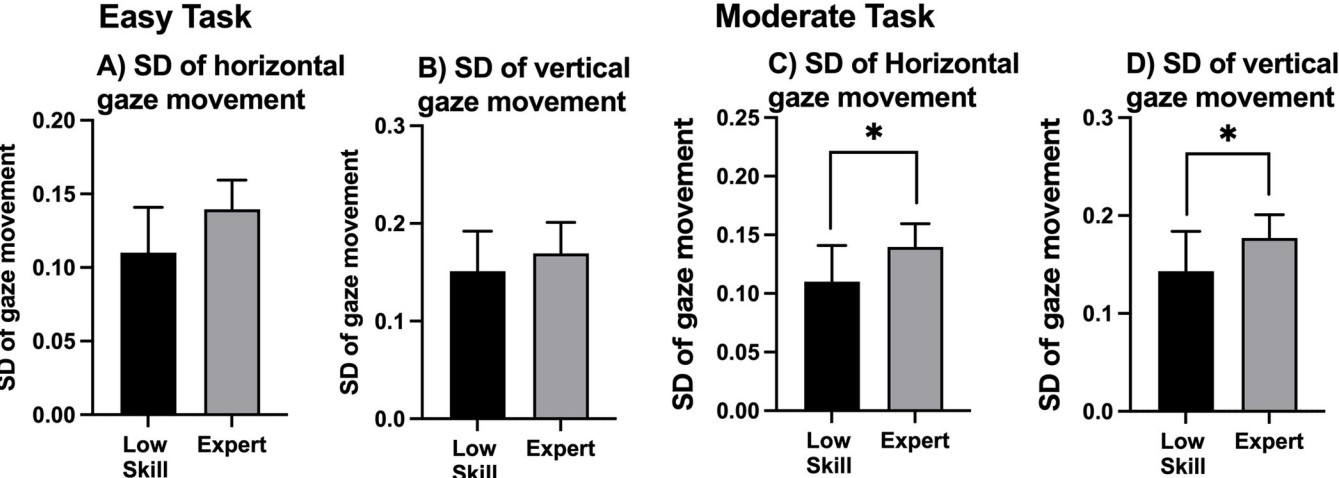

**Fig 4. Standard deviations (SDs) of the horizontal and vertical gaze movements.** The error bars indicate the between-participant SDs of the individual values. The significance level was set at \*$p < .05$.

and the center of the monitor did not exhibit normality or homogeneity. Therefore, these data were analyzed using the non-parametric method (Wilcoxon rank-sum test; expert group vs. low-skill group). The datasets that exhibited normal distribution and homogeneity were tested using the unpaired t-test (expert group vs. low-skill group). Statistical significance was set at $p < .05$.

## Results

### Performance level (kill-to-death ratio)

During the easy task, the unpaired t-test revealed that expert players eliminated more enemies than low-skill players (expert group: 17.2±8.4; low-skill group: 7.4±6.1; t = -3.00; $p < .001$). However, there was no significant difference in the death times of the expert and low-skill players (expert group: 1.6±1.8 minutes; low-skill group: 0.9±0.9 minutes; non-significant difference). During the moderate task, the Wilcoxon rank-sum test found that expert players killed significantly more enemies than the low-skill players (expert group: 10.0±8.1; low-skill group: 1.6±1.8; $p = .001$). Additionally, there was no significant difference in the death times of expert and low-skill players (expert group: 2.5±2.2 minutes; low-skill group: 2.5±1.4 minutes; non-significant difference).

Additionally, the performance level of both groups during the moderate task was significantly lower than that during the easy task (expert group: $p = .05$; low-skill group: $p = .02$).

### Gaze distribution

Fig 3 shows the representative gaze movement on the monitor. Based on this, the expert players had a wider gaze distribution than the low-skill players. However, as shown in Fig 3C and 3D, there was no significant difference in the distance between the center of the monitor and the center of the heatmap of the expert and low-skill players. Fig 4 shows the distribution of gaze movement through the SD of the gaze movement (SD of the horizontal gaze movement and SD of the vertical gaze movement; within-participant SD). During the easy task, there were no significant differences in the SDs of the horizontal and vertical gaze movements of the low-skill players and expert players. Based on the unpaired t-test, the expert players had a significantly wider gaze distribution than the low-skill players during the moderate task (SD of

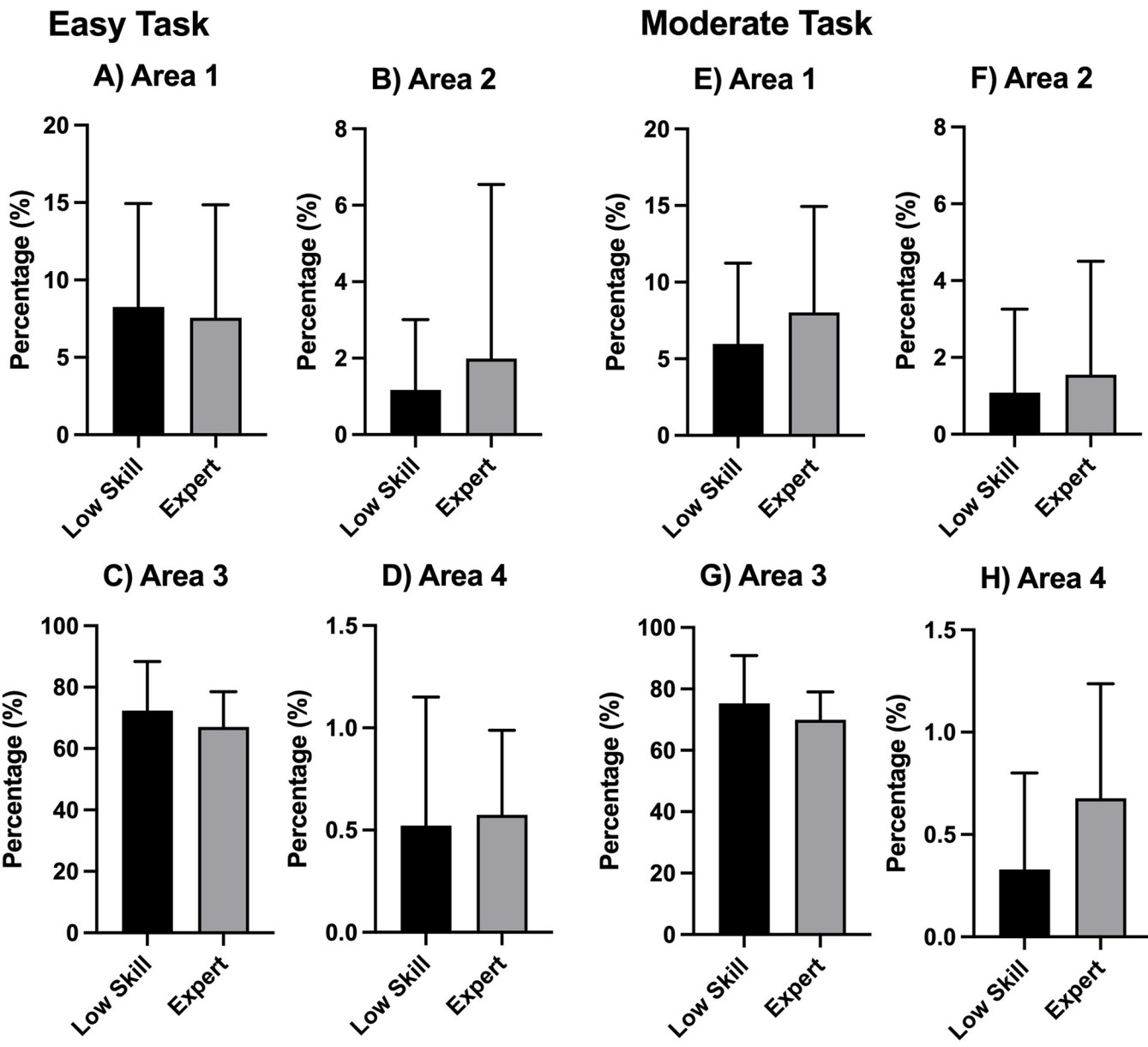

**Fig 5. Areas of interest.** Each graph indicates the percentage of fixation in each area.

the horizontal gaze movement: t = -2.605, *p* = .017; SD of the vertical gaze movement: t = -2.334, *p* = .031).

### Area of interest

A comparison of the percentage values representing fixation in the AOIs showed no significant difference between the low-skill and expert players in all areas (Fig 5).

### Characteristics of fixation

Fig 6 shows the characteristics of fixation. Regarding the fixation duration (Fig 6A and 6D), there was no significant difference between the low-skill and expert players during the easy

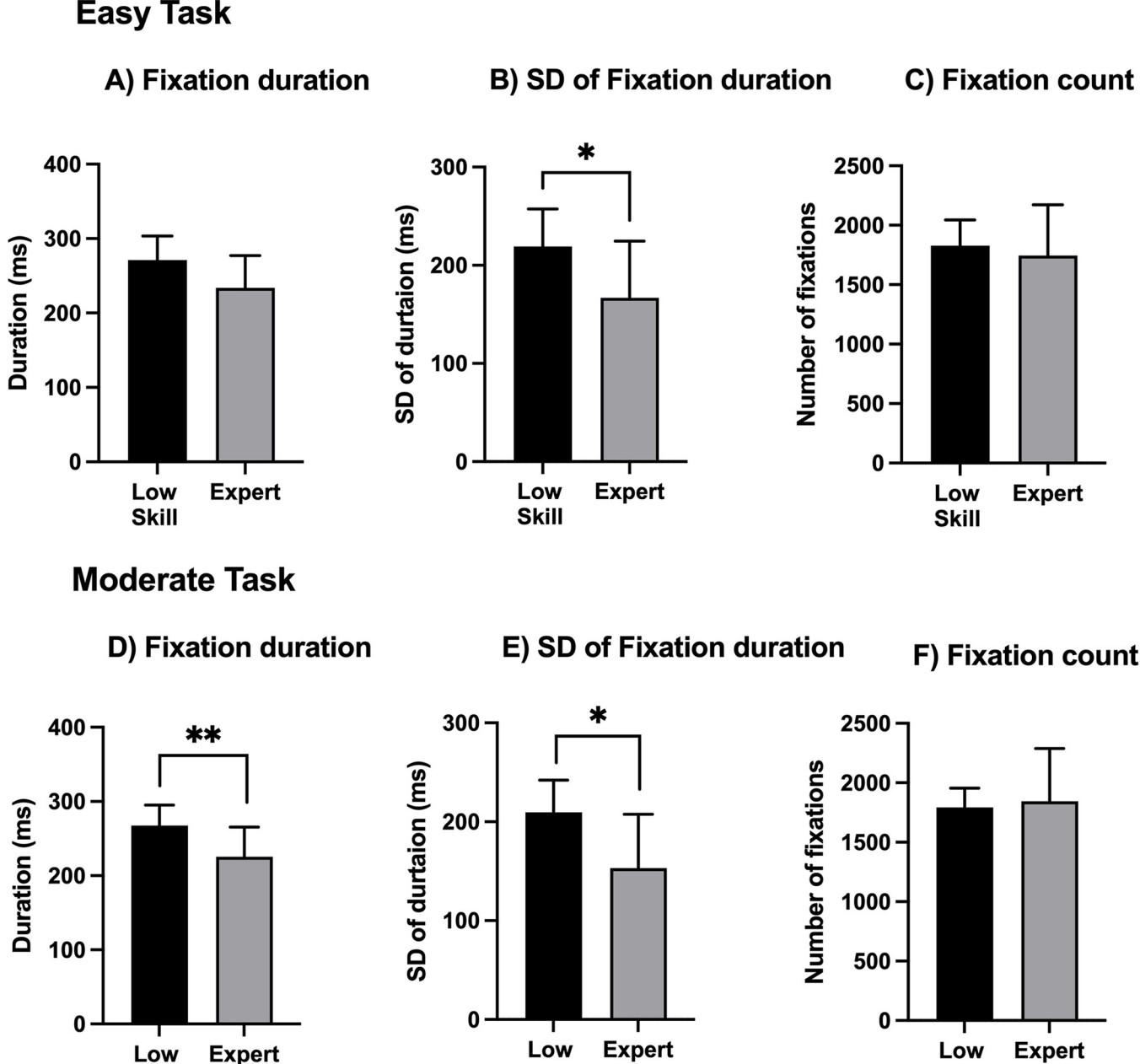

**Fig 6. Characteristics of fixation.** The bar plots show the average fixation duration (A and D). (B and E) The average within-participant SD of the fixation duration. Error bars indicate the between-participant SD of the individual values. The significant levels were set at *$p$ < .05 and **$p$ < .01.

task (based on the Wilcoxon rank-sum test results). However, the Wilcoxon rank-sum test found that during the moderate task, expert players had a shorter fixation duration ($p$ = .007). The SD of each fixation duration is shown in Fig 6B and 6E (within-participant SD). The unpaired t-test revealed that the SD of the fixation duration of expert players during the easy task was significantly shorter than that of the low-skill players (t = 2.327; $p$ = .031). Similarly, the SD of the fixation duration of expert players during the moderate task was significantly shorter than that of the low-skill players (t = 2.724; $p$ = .013). There was no significant

difference in the number of fixations of the expert and low-skill players during the easy and moderate tasks.

## Discussion

This study compared the gaze movement strategy of LoL expert players and low-skill players during specific tasks. The task was conducted using a computer AI system and LoL. Expert players had been playing LoL for a significantly longer time than low-skill players. Additionally, expert players had a significantly superior performance level than low-skill players, and both expert players and low-skill players had lower performance levels during the moderate task than during the easy task. Therefore, expert and low-skill players can be clearly divided into two groups based on their years of experience and performance level. The fact that expert and low-skill players had lower performance levels during the moderate task indicated that each task was properly established. Additionally, because there was no significant difference in the death times of the expert and low-skill players, it could be confirmed that both types of players properly concentrated on the task. The gaze movement data collected during the easy task showed no significant differences between the expert and low-skill players in terms of the SDs of the horizontal and vertical gaze movements, fixation durations, and number of fixations. These results were consistent with those of previous studies that analyzed players who engaged in tasks with a relatively low difficulty level [9]. This suggests that esports, or at least LoL, with low difficulty levels do not require much gaze movement. However, the gaze movement data of the expert players and low-skill players gathered during the moderate task were significantly different. These data showed that the moderate task revealed differences between gaze movements of expert and low-skill players. This suggests that when the difficulty of esports is increased, more gaze movements are required. Moreover, there was no significant difference in the distance between the center of the monitor and the center of the heatmap of the expert and low-skill players. A previous study indicated that the peripheral view is advantageous because it provides for better visual information gathering/processing among soccer players [21]. However, the results of the current study contradict those results [21]. The results of the current study suggest that expert players prioritize accurately checking information using the central view rather than the peripheral view, which is different from the visual information processing skill of general sports players.

A comparison of the SD values of gaze movement showed that expert players had a more widely distributed SD than low-skill players (Fig 4C and 4D). This wide distribution among expert players can be attributed to the characteristics of esports. For example, the information regarding the overall flow of the task (e.g., location and movement of enemy units) was located in the lower right corner of the monitor, whereas the information about the necessary skills for fighting enemies was located at the bottom of the monitor. Therefore, to achieve a higher performance level during the game, players should focus on the entire area of the monitor using wide gaze movement. However, there was no significant difference in the AOI of expert players and low-skill players (Fig 5). There are three possible reasons for this. First, the expert players had significantly more years of experience than low-skill players. Additionally, the expert players exhibited a higher performance level. However, both expert players and low-skill players had played LoL for at least 2 years. Therefore, all participants were already familiar with the LoL interface, thus ensuring that the fixation position of all participants was properly distributed in each area. Second, the AOI was defined as the fixation in each area. Therefore, it is possible that simple gaze movements did not affect the determination of the AOI. Finally, the task that was performed during experiments and the task performed during real LoL competitions have different features. During the task used for the experiment, players could only control

their character in a specific area. However, during real LoL competitions, players have to control their character in a wider area, resulting in the increased importance of the mini-map. Therefore, it is possible that the difference between the experimental task and the real LoL competition affected the results of this study. Regarding the fixation duration, expert players had a shorter fixation duration than low-skill players (Fig 6D). There are two possible reasons for this. First, the expert players had been trained to judge information within a shorter amount of time. Notably, LoL has many multiple object-tracking elements. Through the process of tracking multiple objects, expert players of LoL develop the ability to assess information within a short period of time. Second, expert players can predict what information should be obtained or assessed first. The expert players moved their gaze position after predicting such information. Therefore, expert players do not need to keep their gaze at only one point for a long time. This type of structured scanning performed by expert players was observed during a flight simulator study [22] that found that, compared to novices, professional pilots used more complex and elaborate visual scanning patterns to obtain task-relevant information. The SD of the fixation duration showed that expert players had a smaller SD than low-skill players (Fig 6B–6E). Combining the results of the fixation duration and the SD of the fixation duration showed that expert players were able to consistently obtain information during a short duration. There was no significant difference between expert and low-skill players in terms of the number of fixations (Fig 6C and 6F), which is consistent with the results of previous studies that found no significant difference in the number of fixations of professional esports players and non-professional esports players [2]. Furthermore, it is not important to fix the gaze position many times. It is important to consistently fix the gaze movement for a short time, however.

Regarding general sports, it is well-known that skilled and non-skilled athletes have different gaze movement characteristics. Specifically, skilled athletes have a more efficient gaze movement pattern [23]. For example, during a judo fight scene, skilled judo players will perform fewer fixations and longer fixation durations compared to low-skilled judo players [24]. Judo and LoL have similar characteristics because they require fast reactions; however, the results of the fixation durations remain contradictory. Additionally, during a shooting scene that required dynamic gaze movement, the fixation time of the gaze movement was longer when the target was hit than when the target was not hit [25]. Unlike shooting, LoL requires more information to be processed at the same time; therefore, it is possible that shooting experts had a shorter fixation duration than LoL experts. Two research studies [24, 25] found that the fixation characteristics of LoL expert players differ from those of closed-skills (skills performed in a stable or largely predictable environmental setting) sports players. However, it is known that expert open-skills (skills performed in an unpredictable and constantly changing environment) sports players have short fixation durations [26]. For example, soccer players exhibit fixation during only 2.3% of real soccer competitions [27]. Specifically, soccer players use their gaze movement to search for information during the competition instead of fixing their gaze position. During ice hockey, whenever scanning a wide area is required, expert players exhibit a shorter fixation duration [28].

Regarding the characteristics of the gaze control strategies during closed-skill sports and open-skill sports and those during our study, LoL players exhibited a gaze control strategy similar to that of open-skill sports players.

In general, training methods for sports, including esports, are greatly affected and controlled by gaze movement, and it is well-known that vision training can improve the performance level of players [29–31]. Additionally, previous research used video games as a tool to improve the visual search ability [32]. Further research is necessary to assess training methods using gaze movements for esports players. For example, gaze movement training, which

involves chasing multiple three-dimensional objects, is used to improve performance levels of general sports players [33]. This training method is based on the superior visual attention skills of general athletes [34]. Therefore, it can be useful for developing new training methods for expert esports players based on their characteristics (short gaze fixation time). Additionally, the fixation times measured during specific situations may be useful to esports coaches/ instructors because they can help them understand the cognitive abilities of the players. Esports has many benefits, such as improving cognitive function and providing pleasure. However, mental problems (e.g., insomnia and other mental disorders) caused by playing esports for long periods are becoming social issues [35]. Therefore, through the results of the current study, it is possible to overcome these problems by developing training methods other than simply playing esports that can improve performance.

## Limitations

During this study, we were not able to strictly control the head position of each participant; therefore, this may have affected the reliability of our data. Additionally, the tasks performed during the actual game and the tasks performed by the players during this study had similarities and differences. During the actual game, the purpose is to win. However, during the experimental tasks, the purpose was to eliminate as many enemies as possible. This was the main difference between the experimental task and the real game. However, the similarity between the experimental task and the real game was that participants have to fight the enemy by properly using information and skills. Similar to the experimental task, during the real game, it is advantageous to eliminate as many characters as possible to achieve success. Therefore, we cannot conclude that the findings of this study accurately depict or can be applied to the real-life players of LoL. Moreover, it is necessary to investigate whether the gaze movements of expert players are similar to those of skilled players of other game genres or under other conditions. Finally, we did not observe any influence of sex because all participants were male. Therefore, future studies should recruit male and female participants and conduct an analysis of sex differences in the gaze control ability during esports.

## Conclusion

LoL was used to examine the gaze movement features of MOBA players. Expert players had a wider gaze distribution and a shorter and more consistent fixation duration during the moderate task. These characteristics ultimately enabled them to obtain the necessary information in a wider area within a short time. Furthermore, these specific gaze movements contributed to the high-performance levels of LoL expert players.

## Supporting information

**S1 File. Total gaze movement data of the participants.**
(DOCX)

**S2 File. Details of the G*power protocol.**
(DOCX)

## Acknowledgments

The authors sincerely thank Geon Hwang and Nohyon Park, who gave valuable advice about League of Legends. Additionally, the authors are grateful for the eye-tracker provided by Waseda University.

## Author Contributions

**Conceptualization:** Inhyeok Jeong, Kimitaka Nakazawa.

**Data curation:** Inhyeok Jeong.

**Formal analysis:** Inhyeok Jeong.

**Investigation:** Inhyeok Jeong.

**Methodology:** Inhyeok Jeong.

**Resources:** Inhyeok Jeong.

**Supervision:** Kazutoshi Kudo, Naotusgu Kaneko, Kimitaka Nakazawa.

**Validation:** Naotusgu Kaneko.

**Visualization:** Inhyeok Jeong, Naotusgu Kaneko, Kimitaka Nakazawa.

**Writing – original draft:** Inhyeok Jeong.

**Writing – review & editing:** Inhyeok Jeong, Kazutoshi Kudo, Naotusgu Kaneko, Kimitaka Nakazawa.

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
