## [Decision Letter · Decision Letter 0]

2 May 2023

PONE-D-22-29649Esports experts have a wide gaze distribution and short gaze fixation duration: A focus on League of Legends playersPLOS ONE

Dear Dr. Nakazawa,

Thank you for submitting your manuscript to PLOS ONE. After careful consideration, we feel that it has merit but does not fully meet PLOS ONE’s publication criteria as it currently stands. Therefore, we invite you to submit a revised version of the manuscript that addresses the points raised during the review process. As reviewer 2 stated, the purpose and relevance of the study should be better motivated. Further, implications and conclusions drawn from the study should be more clearly outlined.

We look forward to receiving your revised manuscript.

Kind regards,

Peter Andreas Federolf

Academic Editor

PLOS ONE

 “This research is supported by the JSPS KAKENHI (grant numbers JP18H04082 and JP18HKK0272), JST-Mirai Program (grant number JP20349063), and JST-MOONSHOT program (grant number JPMJMS2012–2-2–2).” 

4. We note that Figures 1 and 2 in your submission contain copyrighted images. All PLOS content is published under the Creative Commons Attribution License (CC BY 4.0), which means that the manuscript, images, and Supporting Information files will be freely available online, and any third party is permitted to access, download, copy, distribute, and use these materials in any way, even commercially, with proper attribution. For more information, see our copyright guidelines: http://journals.plos.org/plosone/s/licenses-and-copyright.

a. You may seek permission from the original copyright holder of Figures 1 and 2 to publish the content specifically under the CC BY 4.0 license.

Reviewers' comments:

Reviewer's Responses to Questions

**Comments to the Author**

1. Is the manuscript technically sound, and do the data support the conclusions?

Reviewer #1: Yes

Reviewer #2: Partly

2. Has the statistical analysis been performed appropriately and rigorously? 

Reviewer #1: Yes

Reviewer #2: Yes

3. Have the authors made all data underlying the findings in their manuscript fully available?

Reviewer #1: Yes

Reviewer #2: Yes

4. Is the manuscript presented in an intelligible fashion and written in standard English?

Reviewer #1: Yes

Reviewer #2: Yes

5. Review Comments to the Author

Reviewer #1: Please elaborate on these points in the revised version:

1) is there an age and gender influence on results? can the authors speculate? or do you have data on? please elaborate.

2) the low versus high skill differentiation can be defined, please add in the method section. It is assumed that the players can gain qualifications, but is that something that might be related to genetics, form and anatomical features that one is born with. Please elaborate.

3) the authors have mentioned the limitations of the study. Can they elaborate further on the internal validity and external validity of this study? i.e., is generalization of data possible or not?

4) please add few lines on use of these data in a practical setting. How the findings can be used for training, coaching, etc. Please elaborate.

Reviewer #2: First of all, I would like to thank the authors for their interesting work. However, I have some major and minor comments, which are addressed below.

In general, it is important to start somewhere in esports research. Still, I had a challenging time to follow the necessity of the current study as it is obvious that expert players have a better visual processing than non-elite players based on existing literature. The argument, why the authors focus on gaze distribution and fixation exactly, was not stated clear enough. Further, what is the message based on the results of the submitted study? Could training in esports benefit from the results or similar? In the current state of the manuscript, implications based on the findings are missing. I would like to encourage the authors to improve their manuscript to become accepted for publication.

Major comments:

Line 80: The authors state that players need to „process the most possible visual information“ and conclude in their hypothesis that „highly skilled LoL players (experts) have a wider gaze distribution and shorter gaze fixation duration compared to LoL players with lower skill levels“. I understand the intention, however, to me this link is a little bit vague as there are various aspects that might lead to a better performance (faster/more accurate motor control, better cognitive processing, etc). I would like to ask the authors to give a better argument why gaze distribution and fixation are important to analyse in the context of esport (LoL) performance.

Line 150: Please add references for validation of the Pupil Labs eye tracker.

Figure 3: When I focus on the heatmap, experts had their gaze way more centred than the non-experts and were able to perform better this way. What about a superior peripheral view of the experts as reason for better performance? This would also refer to the statement in line 300 that „it is possible that simple gaze movements did not affect the determination of the AOI“. Again, what about peripheral view as reason for better visual information gathering/processing (Spierer, D. K., Petersen, R. A., & Duffy, K. (2011). Response time to stimuli in division I soccer players. The Journal of Strength & Conditioning Research, 25(4), 1134-1141.). Did the authors control for this?

Line 345-348: The authors mention training and visual performance. What is the output for coaches and esport athletes based on the results of the present study? Is it possible to train gaze distribution and fixation or will these abilities improve by just playing? I would like to ask the authors to add some more information and references on this topic, as this would support the value of the present paper.

Line 354: „These ultimately enabled them to obtain and analyse information at a faster pace and in a wider area“ did the study design test this statement? I do not agree that gaze distribution and fixation duration automatically lead to e.g. analysis of information.

Line 358: I would highly appreciate if the topic of not controlled peripheral view would be part of the limitations.

Minor comments:

Line 48: Repetition

Line 49: „It is possible that specific gaze movement is a key element of winning the game.“ Is there a reference for this statement? References 11-13 give a hint in this direction.

Line 68-69: Wording

Line 76: A specification of the needed motor control abilities would be helpful for people who are not familiar with LoL and the needed mechanics.

Line 84: methods

Line 92: How many participants had corrected vision and to what degree?

Line 126: Please add „in years“ to the LoL experience

Line 141: Regarding the „higher-level intelligence“, a short explanation how the intelligence increases might be helpful for people who are not familiar with the game.

Line 152: How was that controlled?

Line 155: Version of the Pupil Core software?

Line 160: Was the monitor 24 by 32 inches or could the use their own equipment with various sizes?

Line 188ff: Which tool was used to calculate the mentioned variables?

Line 331: I cannot follow this argument, what do you mean by „therefore, it is natural that the opposite result was obtained for the fixation duration“?

Figure 3: What unit is represented on the axis?

6. PLOS authors have the option to publish the peer review history of their article (what does this mean?). If published, this will include your full peer review and any attached files.

Reviewer #1: No

Reviewer #2: No

---

## [Author Response · Author response to Decision Letter 0]

22 May 2023

Response to Reviewers and Academic Editor

Dear Reviewers and Academic Editor:

Thank you very much for your feedback, comments, and suggestions. Our responses are as follows.

Academic Editor:

Response: We sincerely appreciate your comment. We changed the manuscript style in accordance with PLOS ONE’s style requirements.

 “This research is supported by the JSPS KAKENHI (grant numbers JP18H04082 and JP18HKK0272), JST-Mirai Program (grant number JP20349063), and JST-MOONSHOT program (grant number JPMJMS2012–2-2–2).” 

Response: Thank you for your comments. We added the financial disclosure information in the PLOS ONE Submission system.

Response: We appreciate your comments about ethical issues. Following your instructions, we included the full name of the IRB and information about written consent.

Change: [Page 7-8, Line 116-Line 121] The written consent form included the purpose of the study, methods, privacy policy, and risks of the experiment; the option to withdraw consent was also provided. This research was approved by the Human Research Ethics Committee of the University of Tokyo (Institutional Review Board: Ethical Review Committee for Experimental Research Involving Human Subjects, Graduate School of Arts and Sciences and the College of Arts and Sciences, University of Tokyo).

4. We note that Figures 1 and 2 in your submission contain copyrighted images. All PLOS content is published under the Creative Commons Attribution License (CC BY 4.0), which means that the manuscript, images, and Supporting Information files will be freely available online, and any third party is permitted to access, download, copy, distribute, and use these materials in any way, even commercially, with proper attribution. For more information, see our copyright guidelines: http://journals.plos.org/plosone/s/licenses-and-copyright.

a. You may seek permission from the original copyright holder of Figures 1 and 2 to publish the content specifically under the CC BY 4.0 license.

Response: We appreciate your comments about copyrighted images. We cannot obtain permission from the original copyright holder. Therefore, we replaced the images used as Figures 1 and 2.

Change: Figure 1 and Figure 2.

Figure 1.

Figure 2.

Reviewer #1

1) is there an age and gender influence on results? can the authors speculate? or do you have data on? please elaborate.

Response: We sincerely appreciate your comments. Previous research has indicated that sex differences influence the perception (Kim et al., 2017). However, because there are more male gamers, they are easier to recruit than female gamers. This experiment was conducted among only male gamers. Thus, there are no data available in the current study to examine the effects of sex differences. We believe that the lack of examinations of sex influences is a limitation of the current study. We have added this to the limitation section.

Kim, S. J. (2017). Gender inequality in eSports participation: Examining league of legends (Master’s thesis). University of Texas, Austin. Retrieved from https://repositories.lib. utexas.edu/bitstream/handle/2152/62914/KIM-THESIS-2017.pdf?sequence1⁄41

Change: [Page 25 Line 399-Line 401] Finally, we did not observe any influence of sex because all participants were male. Therefore, future studies should recruit male and female participants and conduct an analysis of sex differences in the gaze control ability during esports.

2) the low versus high skill differentiation can be defined, please add in the method section. It is assumed that the players can gain qualifications, but is that something that might be related to genetics, form and anatomical features that one is born with. Please elaborate.

Response: We appreciate your suggestion. In this section, expert and low-skill players were divided based on their official rank. We did not collect information about participants’ genetics and anatomical features. Previous studies have already claimed that playing video games or esports improves performance (e.g., reaction time) beyond a certain level. However, there is a possibility that visual movement has a genetic limit, such as that for motor skills. Therefore, we will collect this information when planning future research.

3) the authors have mentioned the limitations of the study. Can they elaborate further on the internal validity and external validity of this study? i.e., is generalization of data possible or not?

Response: We sincerely appreciate your comment. We believe that the current study has internal validity. All participants were recruited randomly and performed the same tasks. Moreover, participants were divided into two groups by using the same official ranking system. However, we believe that the results obtained during the current study cannot be readily generalized to other games or tasks, and the gaze movement needs to be verified among more genres of games and situations. Thus, we added the description of external validation in the limitation section. 

Change: [Page 24-25 Line 397-Line 399] Moreover, it is necessary to investigate whether the gaze movements of expert players are similar to those of skilled players of other game genres or under other conditions.

4) please add few lines on use of these data in a practical setting. How the findings can be used for training, coaching, etc. Please elaborate.

Response: Thank you for this helpful comment. We have added specific information about how the results of the current study can be used for esports training.

Change: [Page 23 Line 375-Line 380] For example, gaze movement training, which involves chasing multiple three-dimensional objects, is used to improve performance levels of general sports players [34]. This training method is based on the superior visual attention skills of general athletes [35]. Therefore, it can be useful for developing new training methods for expert esports players based on their characteristics (short gaze fixation time). Additionally, the fixation times measured during specific situations may be useful to esports coaches/instructors because they can help them understand the cognitive abilities of the players. 

Reviewer #2 

First of all, I would like to thank the authors for their interesting work. However, I have some major and minor comments, which are addressed below.

Response: We sincerely appreciate your careful review and helpful comments. We have revised the manuscript according to your comments and opinions. 

In general, it is important to start somewhere in esports research. Still, I had a challenging time to follow the necessity of the current study as it is obvious that expert players have a better visual processing than non-elite players based on existing literature. The argument, why the authors focus on gaze distribution and fixation exactly, was not stated clear enough. Further, what is the message based on the results of the submitted study? Could training in esports benefit from the results or similar? In the current state of the manuscript, implications based on the findings are missing. I would like to encourage the authors to improve their manuscript to become accepted for publication.

Response: Thank you for your comments and advice regarding our study. First, we added the reason for focusing on gaze distribution and fixation in the current study. Next, the benefit of understanding the gaze movement of LoL players was added (e.g., beneficial to training). Finally, the limitations of the current study have been added to the Limitations section. We hope the manuscript is now acceptable.

Change: [Page 5-6 Line 83-Line 85] Confirmation of the information through gaze movement is the first step involved in information processing using cognitive functions [16]. 

Change: [Page 23 Line 375-Line 385] For example, gaze movement training, which involves chasing multiple three-dimensional objects, is used to improve performance levels of general sports players [34]. This training method is based on the superior visual attention skills of general athletes [35]. Therefore, it can be useful for developing new training methods for expert esports players based on their characteristics (short gaze fixation time). Additionally, the fixation times measured during specific situations may be useful to esports coaches/instructors because they can help them understand the cognitive abilities of the players. Esports has many benefits, such as improving cognitive function and providing pleasure. However, mental problems (e.g., insomnia and other mental disorders) caused by playing esports for long periods are becoming social issues [36]. Therefore, through the results of the current study, it is possible to overcome these problems by developing training methods other than simply playing esports that can improve performance.

Change: [Page 24-25 Line 397-Line 401] Moreover, it is necessary to investigate whether the gaze movements of expert players are similar to those of skilled players of other game genres or under other conditions. Finally, we did not observe any influence of sex because all participants were male. Therefore, future studies should recruit male and female participants and conduct an analysis of sex differences in the gaze control ability during esports.

Major comments:

Line 80: The authors state that players need to „process the most possible visual information“ and conclude in their hypothesis that „highly skilled LoL players (experts) have a wider gaze distribution and shorter gaze fixation duration compared to LoL players with lower skill levels“. I understand the intention, however, to me this link is a little bit vague as there are various aspects that might lead to a better performance (faster/more accurate motor control, better cognitive processing, etc). I would like to ask the authors to give a better argument why gaze distribution and fixation are important to analyse in the context of esport (LoL) performance.

Response: We agree with your comments. We added the reason why gaze distribution and short fixation time are important for esports (LoL) performance.

Change: : [Page 5-6 Line 83-Line 85] Confirmation of the information through gaze movement is the first step involved in information processing using cognitive functions [16]. 

Change: [Page 6 Line 88-Line 92] If LoL players can obtain a large amount of visual information provided over a wide area within a short period of time, then they can shorten the information-processing time. This is essential for esports because quick reactions and decisions are required to be successful. During esports, it is beneficial to have shorter information-processing times than those of the opponent players.

Line 150: Please add references for validation of the Pupil Labs eye tracker.

Response： Thank you for your kind concern about the validation of the Pupil-Labs eye tracker. We added the reference for validation of the Pupil-Labs eye tracker.

Change: [Page 11 Line 175] Pupil-Core was validated by the manufacturer [20]. 

Figure 3: When I focus on the heatmap, experts had their gaze way more centred than the non-experts and were able to perform better this way. What about a superior peripheral view of the experts as reason for better performance? This would also refer to the statement in line 300 that „it is possible that simple gaze movements did not affect the determination of the AOI“. Again, what about peripheral view as reason for better visual information gathering/processing (Spierer, D. K., Petersen, R. A., & Duffy, K. (2011). Response time to stimuli in division I soccer players. The Journal of Strength & Conditioning Research, 25(4), 1134-1141.). Did the authors control for this?

Response： Thank you for your comment about Figure 3. As you mentioned, the peripheral view is advantageous for visual information gathering/processing. Thus, we analyzed the distance between the center of the monitor screen and the center of the heatmap. However, there was no significant difference between expert players and low-skill players in terms of these data (Figure 3C and Figure 3D). This suggested the possibility that expert players do not process information based on the peripheral view, unlike the previous study. We added this information in the Results and Discussion sections. 

Change: [Page 14 Line 220-Line 223] The results showed that the performance level of the moderate task, AOI, fixation durations of the easy and moderate tasks, number of fixations of the easy and moderate tasks, and the distance between the center of the heatmap and the center of the monitor did not exhibit normality or homogeneity. 

Change: [Page 16 Line 243-Line 245] However, as shown in Figure 3C and 3D, there was no significant difference in the distance between the center of the monitor and the center of the heatmap of the expert and low-skill players. 

Change: [Page 16-17, Figure 3 legend] Figure 3. Example of the gaze movement and the distance between the center of the monitor screen and the center of the heatmap. (A and B) Examples of gaze movement during the task. Each black dot shows the location of the gaze movement on the monitor. The heatmap shows the density of the gaze position (red: higher density of the gaze position; blue: lower density of the gaze position. (A) Center of the monitor screen. (B) Distance between the center of the heatmap and the center of the monitor. (C) Center of the heatmap. The values of the x-axis and y-axis are calibrated using a number between 0 and 1 based on the actual monitor size. (C and D) The average distances (center of the monitor and center of the heatmap) of the expert and low-skill players. The bar shows the average distance (center of the monitor and center of the heatmap) of expert and low-skill players. The error bars represent the standard deviation (SD) of each value.

Change: [Page 19-20 Line 307-Line 313] Moreover, there was no significant difference in the distance between the center of the monitor and the center of the heatmap of the expert and low-skill players. A previous study indicated that the peripheral view is advantageous because it provides for better visual information gathering/processing among soccer players [22]. However, the results of the current study contradict those results [22]. The results of the current study suggest that expert players prioritize accurately checking information using the central view rather than the peripheral view, which is different from the visual information processing skill of general sports players.

Line 345-348: The authors mention training and visual performance. What is the output for coaches and esport athletes based on the results of the present study? Is it possible to train gaze distribution and fixation or will these abilities improve by just playing? I would like to ask the authors to add some more information and references on this topic, as this would support the value of the present paper.

Response：　We appreciate your comment. The previous study pointed out that esports such as LoL can be a double-edged sword. A positive is that esports can improve the cognitive function. However, a negative is that playing esports for long periods of time can cause mental illnesses such as insomnia and mood disorders (Wattanapisit et al., 2020). Therefore, we believe that the known negative effects (insomnia and mood disorders, etc.) can be overcome by developing training methods other than simply playing esports. Information about the benefits of developing new training methods based on the characteristics of LoL players’ gaze movement characteristics was added to the Discussion section.

Change: [Page 23-24 Line 379-385] Additionally, the fixation times measured during specific situations may be useful to esports coaches/instructors because they can help them understand the cognitive abilities of the players. Esports has many benefits, such as improving cognitive function and providing pleasure. However, mental problems (e.g., insomnia and other mental disorders) caused by playing esports for long periods are becoming social issues [36]. Therefore, through the results of the current study, it is possible to overcome these problems by developing training methods other than simply playing esports that can improve performance.

Line 354: „These ultimately enabled them to obtain and analyse information at a faster pace and in a wider area“ did the study design test this statement? I do not agree that gaze distribution and fixation duration automatically lead to e.g. analysis of information.

Response： Thank you for your kind comments. We agree with your statement. Therefore, we have reworked this statement.

Change: [Page 25 Line 404-406] These characteristics ultimately enabled them to obtain the necessary information in a wider area within a short time.

Line 358: I would highly appreciate if the topic of not controlled peripheral view would be part of the limitations.

Response： We appreciate your suggestion. We added the additional results and discussion about the peripheral view to the Results and Discussion sections.

Change: [Page 19-20 Line 307-Line 313] Moreover, there was no significant difference in the distance between the center of the monitor and the center of the heatmap of the expert and low-skill players. A previous study indicated that the peripheral view is advantageous because it provides for better visual information gathering/processing among soccer players [22]. However, the results of the current study contradict those results [22]. The results of the current study suggest that expert players prioritize accurately checking information using the central view rather than the peripheral view, which is different from the visual information processing skill of general sports players.

Minor comments:

Response: Thank you for your minor correction of this section. These minor issues have been fixed.

Line 48: Repetition

Response： We apologize for this repetition. The statement on Line 48 has been removed.

Line 49: „It is possible that specific gaze movement is a key element of winning the game.“ Is there a reference for this statement? References 11-13 give a hint in this direction.

Response： Thank you for your comment about this statement. We have added references 11-13 to this statement. 

Change: [Page 4 Line 52-53] It is possible that specific gaze movement is a key element involved in winning the game [9–11].

Line 68-69: Wording

Response： We apologize for using ambiguous wording in this statement. The statement on this line has been corrected.

Change: [Page 5 Line 71-Line 74] In other words, analyses of gaze movement characteristics, especially fixation, can be helpful to understanding gaze movement of esports players and can help determine the origins of the high-performance ability of esports experts. 

Line 76: A specification of the needed motor control abilities would be helpful for people who are not familiar with LoL and the needed mechanics.

Response： We sincerely appreciate your comment. We described in detail the high motor control ability required by LoL players.

Change: [Page 5 Line 80-Line 82] Specifically, during LoL, players are required to have good motor control ability to accurately operate a mouse while pressing the appropriate button on a keyboard as quickly as possible. 

Line 84: methods

Response： We apologize for using the wrong word. “methods” has been changed to “Methods”.

Change: [Page 6 Line 96] Methods

Line 92: How many participants had corrected vision and to what degree?

Response： Thank you for your comment about the participants' vision. Honestly, we did not measure the actual degree of vision of the participants. However, all participants reported that they did not have poor vision when playing the task. This information has been added.

Change: [Page 7 Line 104-105] All participants who performed the task had normal or corrected-to-normal vision.

Line 126: Please add „in years“ to the LoL experience

Response： Thank you. We added the units to the LoL experience.

Change: [Table 1] LoL experience, yr

Line 141: Regarding the „higher-level intelligence“, a short explanation how the intelligence increases might be helpful for people who are not familiar with the game.

Response： We sincerely appreciate your suggestion and agree. Therefore, we added information and an explanation of higher-level intelligence in this statement.

Change: [Page 10 Line 153-Line 155] During the moderate task, the AI system collected gold 1.5-times faster than during the easy task, and skills were used more frequently. 

Line 152: How was that controlled?

Response： Thank you for your comment. We asked the participants to maintain the same head position as that during the start of the task. 

Change: [Page 10 Line 165-166] Then, we asked the participants to maintain the same head position that they used during the start of the task. 

Line 155: Version of the Pupil Core software?

Response： We apologize for the lack of information. We added the specific version of Pupil-Capture.

Change: [Page 11 Line 168-169] During the task, gaze movement was recorded using Pupil-Capture software (version 3.5; https://github.com/pupil-labs/pupil/releases/tag/v3.5). 

Line 160: Was the monitor 24 by 32 inches or could the use their own equipment with various sizes?

Response： Thank you for your comment. Yes, all participants used their own equipment of various sizes. After collecting the data, they were calibrated according to the actual size of the monitor. 

Change: [Page 12 Line 179-181] After calibration, four different markers (width × height: 4 cm × 4 cm) were attached to the monitor (Figure 2A) to normalize the gaze movement, which was quantified using a value between 0 and 1 based on the actual monitor size. 

Line 188ff: Which tool was used to calculate the mentioned variables?

Response： Thank you for your question about the variables on Line 188. The location of the AOI in each area was set through the baseline information of the task used during the current study.

Change: [Page 13 Line 201-Line 202] The location of each fixation for each area was designated as an area of interest (AOI) based on the task interface used during the current experiment (Figure 1). 

Line 331: I cannot follow this argument, what do you mean by „therefore, it is natural that the opposite result was obtained for the fixation duration“?

Response： We apologize for using an ambiguous statement. It compares the characteristics of gaze movement used during general sports (in this part, shooting) and LoL. It corresponds to the claim that different results occurred because LOL requires more dynamic visual information processing than actual shooting. We have edited this statement for clarity.

Change: [Page 22 Line 358-Line 359] Unlike shooting, LoL requires more information to be processed at the same time; therefore, it is possible that shooting experts had a shorter fixation duration than LoL experts. 

Figure 3: What unit is represented on the axis?

Response： We apologize for the missing axis information in Figure 3. The values of the x-axis and y-axis were calibrated with a number between 0 and 1 based on the actual monitor size. This information has been added in the legend of Figure 3.

Change: [Page 16-17 Line 258- 260] The values of the x-axis and y-axis are calibrated using a number between 0 and 1 based on the actual monitor size. (C and D) The average distances (center of the monitor and center of the heatmap) of the expert and low-skill players.

---

## [Decision Letter · Decision Letter 1]

5 Jul 2023

Esports experts have a wide gaze distribution and short gaze fixation duration: A focus on League of Legends players

PONE-D-22-29649R1

Dear Dr. Nakazawa,

We’re pleased to inform you that your manuscript has been judged scientifically suitable for publication and will be formally accepted for publication once it meets all outstanding technical requirements.

Kind regards,

Peter Andreas Federolf

Academic Editor

PLOS ONE

Additional Editor Comments (optional):

Reviewers' comments:

Reviewer's Responses to Questions

**Comments to the Author**

1. If the authors have adequately addressed your comments raised in a previous round of review and you feel that this manuscript is now acceptable for publication, you may indicate that here to bypass the “Comments to the Author” section, enter your conflict of interest statement in the “Confidential to Editor” section, and submit your "Accept" recommendation.

Reviewer #1: All comments have been addressed

Reviewer #2: All comments have been addressed

2. Is the manuscript technically sound, and do the data support the conclusions?

Reviewer #1: Yes

Reviewer #2: Yes

3. Has the statistical analysis been performed appropriately and rigorously? 

Reviewer #1: Yes

Reviewer #2: Yes

4. Have the authors made all data underlying the findings in their manuscript fully available?

Reviewer #1: Yes

Reviewer #2: Yes

5. Is the manuscript presented in an intelligible fashion and written in standard English?

Reviewer #1: Yes

Reviewer #2: Yes

6. Review Comments to the Author

Reviewer #1: The authors have addressed the points raised by this reviewer and accordingly revised the manuscript. There are no more comments.

Reviewer #2: Congratulations to the authors for successfully implementing all suggested comments. In my opinion, the authors could improve their manuscript significantly and the motivation for the conducted study is now easier to follow. I would recommend to accept the manuscript in its current form.

7. PLOS authors have the option to publish the peer review history of their article (what does this mean?). If published, this will include your full peer review and any attached files.

Reviewer #1: No

Reviewer #2: **Yes: **Felix Wachholz

---

## [Editor Report · Acceptance letter]

18 Aug 2023

PONE-D-22-29649R1 

Esports experts have a wide gaze distribution and short gaze fixation duration: A focus on League of Legends players 

Dear Dr. Nakazawa:

I'm pleased to inform you that your manuscript has been deemed suitable for publication in PLOS ONE. Congratulations! Your manuscript is now with our production department. 

Kind regards, 

on behalf of

Dr. Peter Andreas Federolf 

Academic Editor

PLOS ONE